# Nationwide participation in FIT-based colorectal cancer screening in Denmark during the COVID-19 pandemic: An observational study

**Tina Bech Olesen[1]\*, Henry Jensen[1], Henrik Møller[1], Jens Winther Jensen[1], Berit Andersen[2,3], Morten Rasmussen[4,5]**

[1]The Danish Clinical Quality Program – National Clinical Registries (RKKP), Aarhus, Denmark; [2]University Research Clinic for Cancer Screening, Department of Public Health Programmes, Randers Regional Hospital, Aarhus, Denmark; [3]Department of Clinical Medicine, Aarhus University, Aarhus, Denmark; [4]The Colorectal Cancer Screening Programme, the Capital Region, Copenhagen, Denmark; [5]Abdominal Center, Bispebjerg Hospital, Copenhagen, Denmark

**\*For correspondence:**
forthemanuscripts@gmail.com

**Competing interest:** The authors declare that no competing interests exist.

## Abstract

**Background:** Worldwide, most colorectal cancer screening programmes were paused at the start of the COVID-19 pandemic, while the Danish faecal immunochemical test (FIT)-based programme continued without pausing. We examined colorectal cancer screening participation and compliance with subsequent colonoscopy in Denmark throughout the pandemic.

**Methods:** We used data from the Danish Colorectal Cancer Screening Database among individuals aged 50–74 years old invited to participate in colorectal cancer screening from 2018 to 2021 combined with population-wide registries. Using a generalised linear model, we estimated prevalence ratios (PRs) and 95% confidence intervals (CIs) of colorectal cancer screening participation within 90 days since invitation and compliance with colonoscopy within 60 days since a positive FIT test during the pandemic in comparison with the previous years adjusting for age, month and year of invitation.

**Results:** Altogether, 3,133,947 invitations were sent out to 1,928,725 individuals and there were 94,373 positive FIT tests (in 92,848 individuals) during the study period. Before the pandemic, 60.7% participated in screening within 90 days. A minor reduction in participation was observed at the start of the pandemic (PR = 0.95; 95% CI: 0.94–0.96 in pre-lockdown and PR = 0.85; 95% CI: 0.85–0.86 in first lockdown) corresponding to a participation rate of 54.9% during pre-lockdown and 53.0% during first lockdown. This was followed by a 5–10% increased participation in screening corresponding to a participation rate of up to 64.9%. The largest increase in participation was observed among 55–59 years old and among immigrants. The compliance with colonoscopy within 60 days was 89.9% before the pandemic. A slight reduction was observed during first lockdown (PR = 0.96; 95% CI: 0.93–0.98), where after it resumed to normal levels.

**Conclusions:** Participation in the Danish FIT-based colorectal cancer screening programme and subsequent compliance to colonoscopy after a positive FIT result was only slightly affected by the COVID-19 pandemic.

**Funding:** The study was funded by the Danish Cancer Society Scientific Committee (Grant number R321-A17417) and the Danish regions.

## Editor's evaluation

The authors convincingly demonstrate that, in the absence of any shutdowns, the Danish colorectal cancer screening program experienced only minor decreases in program participation during the COVID-19 pandemic period. This likely ensured ongoing program effectiveness in detecting early colorectal cancers and precancerous polyps. The evidence is solid and may serve as guidance for other countries when facing similar public health threats in the future.

## Introduction

The COVID-19 pandemic has impacted the society and the healthcare systems worldwide considerably. In efforts to mitigate the impact of the COVID-19 pandemic on the healthcare system and to minimise the spread of the infection, population-wide restrictions (lockdown) were imposed worldwide. Large parts of the society were closed down and, within the healthcare system, elective procedures were cancelled or postponed and resources were reallocated to take care of patients in need of hospitalisation because of COVID-19.

As a result of the re-organisations within the healthcare systems, the cancer screening programmes were, in most countries, paused at the start of the pandemic (*Morris et al., 2021*; *Dinmohamed et al., 2020*; *Kortlever et al., 2021*; *Vives et al., 2022*; *Walker et al., 2021*). In Denmark, however, the cancer screening programmes including the faecal immunochemical test (FIT)-based colorectal cancer screening programme using faecal samples obtained at home continued throughout the pandemic. Results from other European countries using FIT-based screening programmes have shown that alterations to the colorectal cancer screening programme at the start of the pandemic led to large reductions in the number of people referred, diagnosed and treated for colorectal cancer at the start of the pandemic (*Morris et al., 2021*; *Dinmohamed et al., 2020*) and to reduced participation in screening and screening-derived colonoscopy (*Kortlever et al., 2021*) and longer time interval from a positive screening test to colonoscopy (*Vives et al., 2022*). A study from Canada also found marked reductions in the colorectal cancer faecal test volumes at the start of the pandemic (*Walker et al., 2021*) resulting from a suspension of the FIT-based screening programme. Moreover, it is estimated that the disruptions to the FIT-based colorectal cancer screening programme would result in additional colorectal cancer diagnoses (*de Jonge et al., 2021*). The participation in colorectal cancer screening in Denmark throughout the pandemic has not yet been described–however, one study has shown a 24% reduction in the number of colon cancers diagnosed at the start of the pandemic in Denmark (*Skovlund et al., 2022*) indicating that either the general health-seeking behaviour or the participation in colorectal cancer screening may have changed at the start of the pandemic.

It is well known that social inequities exist across the entire colorectal cancer screening pathway. For example, studies have shown that younger individuals, immigrants, individuals living alone and individuals with a lower income are less likely to participate in colorectal cancer screening (*Larsen et al., 2017*; *Pallesen et al., 2021*). Furthermore, the compliance with colonoscopy is lower among older patients and among patients with underlying disease (*Thomsen et al., 2018*), among immigrants and among individuals living alone (*Pallesen et al., 2021*). A concern is that these social inequities in colorectal cancer screening participation may have been exacerbated during the pandemic.

We examined the colorectal cancer screening participation and compliance with subsequent colonoscopy during the COVID-19 pandemic in Denmark compared with the previous years. Furthermore, we examined whether the participation in colorectal cancer screening and compliance with screen-derived colonoscopy during the COVID-19 pandemic differed across population sub-groups.

## Methods
### Setting

The study was conducted in Denmark, which has a population of approximately 5.8 million inhabitants (*Statistics Denmark, 2021*). All residents in Denmark are eligible for tax-supported health care provided by the Danish government. Nationwide population-based registries in Denmark record extensive administrative and medical data of the whole population, which can be linked using the

unique personal identifier, that is assigned to all residents at birth or immigration (**Schmidt et al., 2014–Schmidt et al., 2019**).

## The colorectal cancer screening programme

In Denmark, screening for colorectal cancer was implemented in 2014 and is offered free-of-charge every 2 years to all individuals aged 50–74 years old living in Denmark. The test is a home-based test, which is mailed directly together with an invitation letter to all invitees. The screening procedure is based on a single-sample FIT (OC Sensor; Eiken Chemical Company, Tokyo, Japan), which can detect invisible amounts of blood in stool samples, which may be associated with bleeding lesions from precancerous adenomas or colorectal cancer at early stages of the disease (**Hewitson et al., 2008**; **Garborg et al., 2013**). Non-participants to screening receive a reminder after 6 weeks.

All individuals with a positive FIT test (≥100 μg haemoglobin/L faeces) receive an invitation for a colonoscopy with a pre-booked time for appointment within 14 days after the positive screening result. Non-participants to colonoscopy are contacted by the administrative regions.

## The COVID-19 pandemic in Denmark

In Denmark, three main waves of the COVID-19 pandemic have occurred; in the spring of 2020, in the winter of 2020/2021 and again in the winter of 2021/2022 (**Statens Serum Institut, 2021a**).

The pandemic response included population-wide restrictions (lockdowns), COVID-19 testing and COVID-19 vaccination. During the lockdowns large parts of the society were closed down and people were advised to stay at home if possible. Large-scale COVID-19 testing was provided free-of-charge to all inhabitants since May 2020 (**Pottegård et al., 2020**). COVID-19 vaccination began in December 2020 and by March 2022, approximately 81% of the population had received two doses and more than 61% had received three doses of the vaccine (**Statens Serum Institut, 2021b**). The

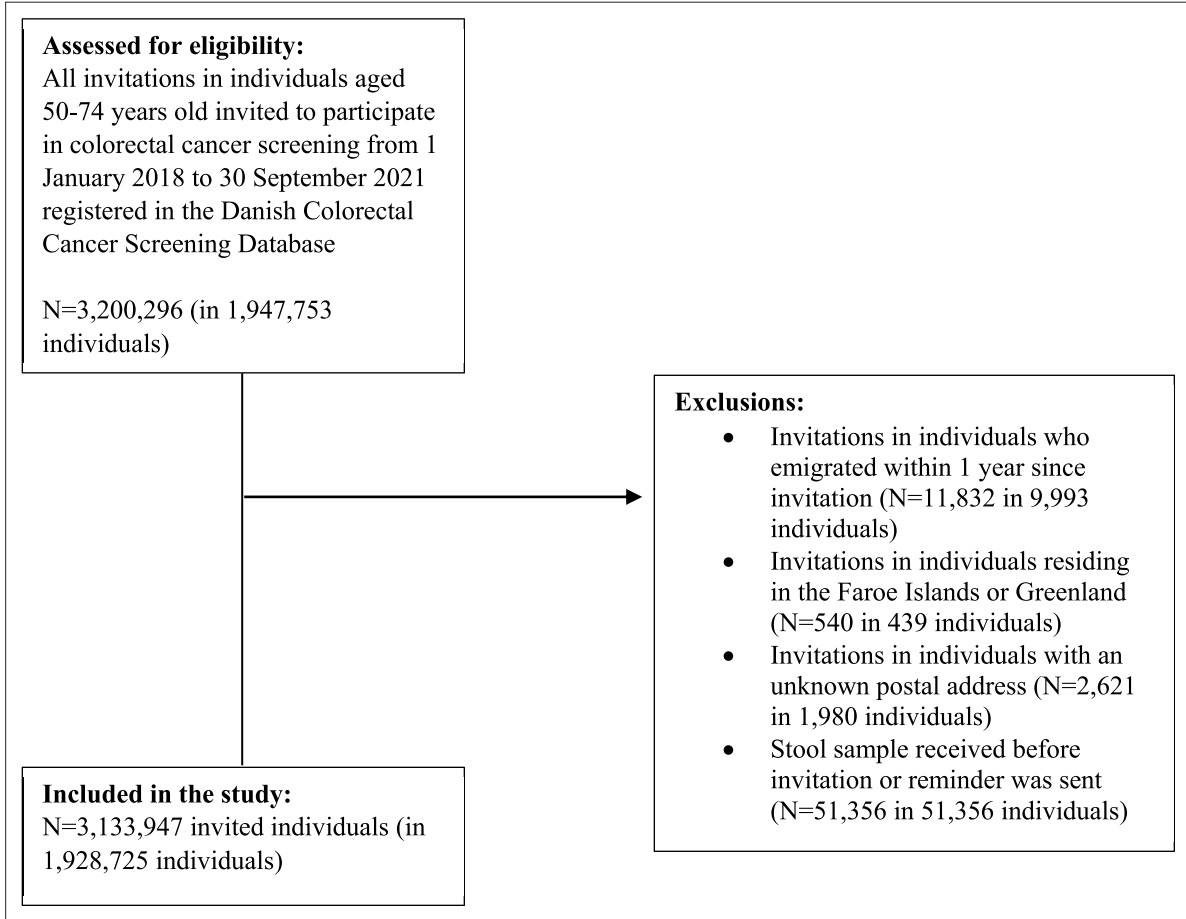

**Figure 1.** Flow-chart of the study population (participation in colorectal cancer screening).

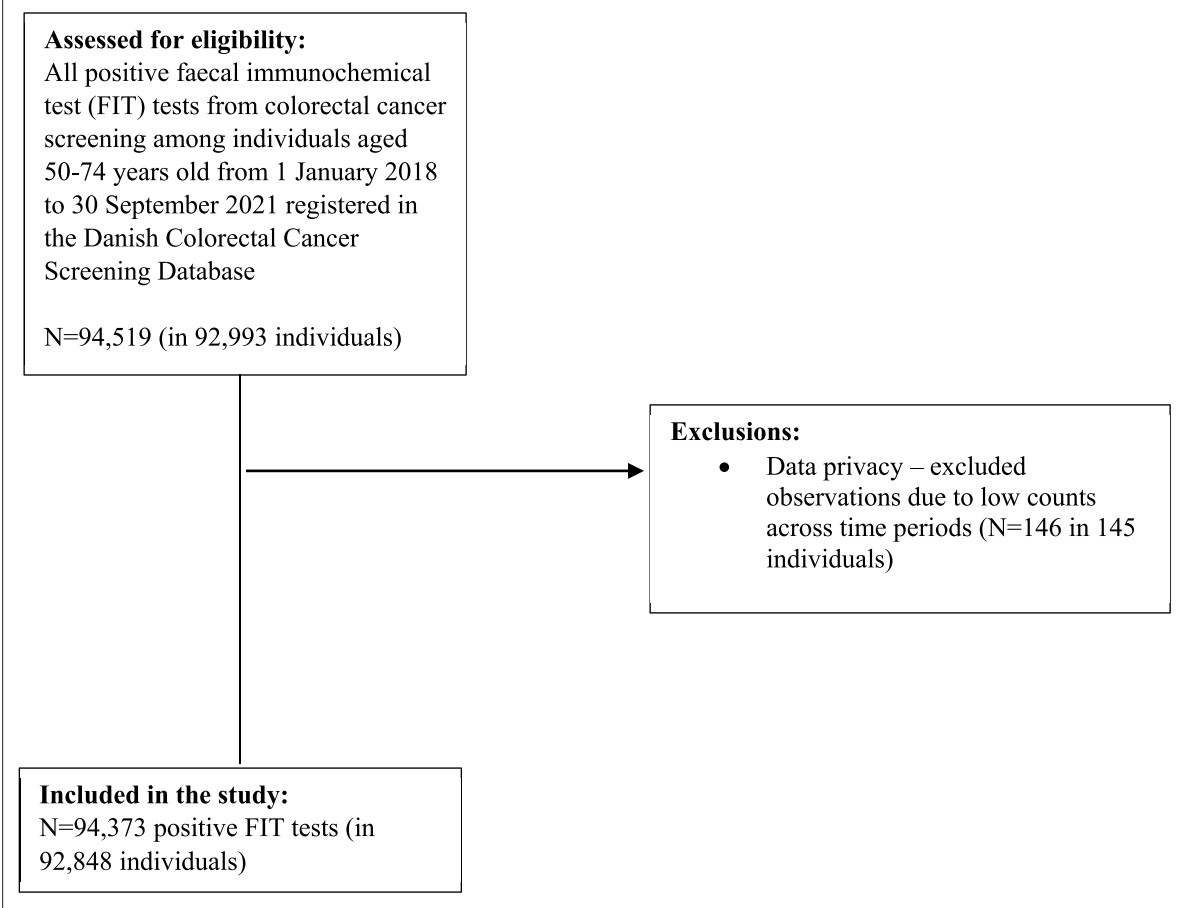

**Figure 2.** Flow-chart of the study population (compliance with colonoscopy).

vaccination strategy comprised vaccinating individuals living in nursing homes first, thereafter individuals ≥85 years, then healthcare personnel, thereafter individuals with underlying health conditions and their relatives and finally, individuals were offered the COVID-19 vaccination by decreasing age (75–79, 65–74, 60–64 years, etc.) (*Sundhedsstyrelsen, 2021*).

## Study population

The study population comprised all invitations in individuals aged 50–74 years old invited to participate in colorectal cancer screening from 1 January 2018 to 30 September 2021, as registered in the Danish Colorectal Cancer Screening Database (*Thomsen et al., 2017*), which contain information on all individuals in Denmark invited to participate in colorectal cancer screening.

To examine participation in colorectal cancer screening, we excluded invitations in individuals who emigrated within 1 year since invitation (N=11,832), invitations in individuals residing in the Faroe Islands or Greenland (N=540), invitations in individuals with an unknown postal address (N=2621) and registrations of stool samples received before an invitation or reminder was sent out (*Figure 1*).

To examine compliance with colonoscopy, we included all positive FIT tests from colorectal cancer screening among individuals aged 50–74 years old from 1 January 2018 to 30 September 2021. In all, 146 positive FIT tests were excluded due to low counts across time periods (*Figure 2*).

## Exposure of interest

The COVID-19 pandemic is the exposure of interest. We defined the different phases of the pandemic in Denmark in accordance with the governmental responses to the COVID-19 pandemic in Denmark, as follows:

- Pre-pandemic period: 1 January 2018 to 31 January 2020

- Pre-lockdown period: 1 February to 10 March 2020
- 1st lockdown: 11 March to 15 April 2020
- 1st re-opening: 16 April to 15 December 2020
- 2nd lockdown: 16 December 2020 to 27 February 2021
- 2nd re-opening: 28 February 2021 to 30 September 2021 (end of inclusion period)

Pre-lockdown and first lockdown was termed 'the start of the pandemic' in this study.

The above-mentioned time periods refer to the time of invitation for colorectal cancer screening and the time of a positive FIT result for each of the outcomes of interest.

## Outcome of interests

The two main outcomes of interests were colorectal cancer screening participation within 90 days since invitation and compliance with colonoscopy within 60 days since a positive FIT result. Further, we evaluated participation within 180 and 365 days since invitation and compliance with colonoscopy within 365 days since a positive FIT result. We thus calculated the proportion of individuals participating in colorectal cancer screening within 90, 180 and 365 days since invitation and the compliance with colonoscopy within 60 and 365 days since a positive FIT result.

## Explanatory variables

The following variables were examined independently: age, sex, ethnicity, cohabitation status, educational level, disposable income and healthcare usage. Age was defined at the date of invitation, as registered in the Danish Colorectal Cancer Screening Database (*Thomsen et al., 2017*). From *Statistics Denmark, 2021*, we obtained information on ethnicity, cohabitation status, educational level and level of income. Ethnicity was categorised as Danish descent, Western immigrant, Non-western immigrant and descendants of immigrants. Cohabitation status was categorised as living alone, cohabiting/co-living, and married (i.e., married or registered partnership) in accordance with *Statistics Denmark, 2021*. Educational level was defined in accordance with the International Standard Classification of Education (ISCED) of the United Nations Education, Scientific and Cultural Organization (UNESCO). We thus categorised level of education into short (ISCED levels 1–2: primary education to upper secondary education), medium (ISCED levels 3–5: vocational education and training to vocational bachelors educations) and long (ISCED levels 6–8: bachelors programmes to PhD programmes) (*Statistics Denmark, 2021*). Income was defined as official disposable income depreciated to 2015 level and categorised into five quintiles. To indicate the level of healthcare use by each patient, we counted the total number of contacts (comprising face-to-face, telephone and e-mail consultations) to general practitioners, private practising medical specialists, physiotherapists and chiropractors in the year for invitation as registered in the Danish National Health Service Register (*Andersen et al., 2011*), which contain information on visits to primary healthcare (e.g., general practitioners and medical specialists) in Denmark since 1990. We categorised healthcare usage into five quintiles of the data as rare (0–3 visits per year), low (4–6 visits per year), average (7–11 visits per year), high (12–18 visits per year) and frequent (≥19 visits per year).

Information on cohabitation status was only available from Statistics Denmark until the end of February 2021, whereas all other socioeconomic variables were available until end of the study period.

## Statistical analyses

We examined characteristics of persons invited to participate in colorectal cancer screening and characteristics of persons with a positive FIT test during the study period. Thereafter, we examined the participation in colorectal cancer screening within 90, 180 and 365 days since invitation overall and stratifying by the explanatory variables per month and during the pandemic phases. Similarly, we examined compliance with colonoscopy within 60 and 365 days since a positive FIT test overall and stratifying by the explanatory variables per month and during the pandemic phases. We also examined the median number of days and interdecentile interval (IDI) from invitation to participation overall and during the different phases of the pandemic, among individuals eventually participating in the screening programme.

Using a generalised linear model (GLM) with log link for the Poisson family with robust standard errors (SEs), we estimated prevalence ratios (PRs) and 95% confidence intervals (CIs) of participation in colorectal cancer screening within 90, 180 and 365 days since invitation among persons invited to

participate in colorectal cancer screening and compliance with colonoscopy within 60 and 365 days since a positive FIT test during the different phases of the pandemic overall and stratifying by the explanatory variables. First, we calculated unadjusted analyses. Thereafter, the analyses were adjusted for month of invitation to allow for seasonality and year of invitation to take into account the annual change in colorectal cancer screening participation. Finally, the analyses were adjusted for age to take into account the effect of age on the other explanatory variables.

We performed a sensitivity analysis to take into account an IT-error that occurred in the spring of 2020 resulting in a reduction in the number of invitations sent out in weeks 11–14 2020 (Central Denmark Region in weeks 11–14 2020, Northern Denmark Region in weeks 12–14 2020 and the rest of Denmark in weeks 13–14 2020). The error meant, that only individuals entering or leaving the screening programme were invited. Thus, during the period with the IT-error only 50 years olds entering the programme, individuals entering the country from abroad, and 73–74 years old leaving the programme were invited. We re-ran the analyses for this by introducing a dummy variable expressing the IT-error in the GLM model.

All analyses were conducted using STATA version 17.0.

## Results

Altogether, 3,133,947 invitations were sent out to 1,928,725 individuals during the study period. Among those 50.5% were women and the median age was 60 years (IQI = 54–67), the majority were of Danish descent (91.4%), most were married (59.4%) and 56% had a short educational level. The distribution of the descriptive characteristics was similar across the pandemic phases (*Table 1*).

### Participation during the COVID-19 pandemic

Before the pandemic, 60.7% participated in colorectal cancer screening within 90 days since invitation (*Figure 3* and *Supplementary file 1*). The results were similar extending the length of follow-up time to 180 and 365 days (data not shown).

A reduction in screening participation within 90 days occurred during February and March 2020 (*Figure 1*) reflected in a PR of 0.95 (95% CI: 0.94–0.96) during pre-lockdown and a PR of 0.85 (95% CI: 0.85–0.86) during first lockdown (*Table 2*). This reduction corresponded to an overall participation rate of 54.9% during pre-lockdown and 53.3% during first lockdown (*Supplementary file 1*). Subsequently, an increase in screening participation was observed (*Figure 1*) reflected in overall PRs of 1.04 (95% CI: 1.04–1.05), 1.09 (95% CI: 1.09–1.10) and 1.11 (95% CI: 1.10–1.12) during first re-opening, second lockdown and second re-opening, respectively (*Table 2*). These increases corresponded to participation rates of 62.4% during first re-opening, 63.0% during second lockdown and 64.9% during second re-opening (*Supplementary file 1*). These estimates were similar when extending the length of follow-up time to 180 and 365 days (data not shown).

### Participation during the COVID-19 pandemic according to socio-economic variables

Throughout the study period, the participation in colorectal cancer screening was lowest among the youngest age group, among men, among immigrants, among individuals living alone or cohabiting, among individuals with a low educational level, a low income and among individuals who rarely use the healthcare system (*Supplementary file 1*). During first lockdown, women, 70–74 years old and individuals with a low income had the lowest participation in screening. From first re-opening and onwards, the largest relative increases in participation was observed among 55–59 years old and among immigrants (*Table 2*).

### Participation among first-time invitees

Altogether, 8.8% (N=276,495) of the study population were first-time invitees. The median age of first-time invitees was 50 years old (IQI: 50–50), 15% were immigrants and 33% rarely used the primary healthcare system (*Supplementary file 2*). Before the pandemic, 53% of first-time invitees participated in screening within 90 days, 54% within 180 days and 55% within 365 days (data not shown). A slight reduction in participation within 90 days was observed during pre-lockdown (PR = 0.95; 95% CI: 0.93–0.98), an increase in participation was found during first lockdown (PR = 1.06; 95% CI:

**Table 1.** Baseline characteristics of people invited to participate in colorectal cancer screening in Denmark from 2018 to 2021.

| | Pre-pandemic (1 January 2018 to 31 January 2020) | Pre-lockdown (1 February 2020 to 10 March 2020) | 1st lockdown (11 March 2020 to 15 April 2020) | 1st re-opening (16 April 2020 to 15 December 2020) | 2nd lockdown (16 December 2020 to 27 February 2021) | 2nd re-opening (28 February 2021 to 30 September 2021) | Total |
|---|---|---|---|---|---|---|---|
| | N (%) | N (%) | N (%) | N (%) | N (%) | N (%) | N (%) |
| Total | 1,804,770 (57.6) | 104,892 (3.3) | 40,146 (1.3) | 562,124 (17.9) | 176,469 (5.6) | 445,546 (14.2) | 3,133,947 (100.0) |
| **Sex** | | | | | | | |
| Men | 893,009 (49.5) | 52,523 (50.1) | 19,285 (48.0) | 276,731 (49.2) | 87,667 (49.7) | 218,713 (49.1) | 1,547,928 (49.4) |
| Women | 911,761 (50.5) | 52,369 (49.9) | 20,861 (52.0) | 285,393 (50.8) | 88,802 (50.3) | 226,833 (50.9) | 1,586,019 (50.6) |
| **Age at invitation** | | | | | | | |
| 50–54 years | 478,804 (26.5) | 14,982 (14.3) | 7322 (18.2) | 127,306 (22.6) | 49,217 (27.9) | 138,988 (31.2) | 816,619 (26.1) |
| 55–59 years | 375,309 (20.8) | 38,988 (37.2) | 5739 (14.3) | 150,615 (26.8) | 32,217 (18.3) | 61,318 (13.8) | 664,186 (21.2) |
| 60–64 years | 332,766 (18.4) | 18,089 (17.2) | 2929 (7.3) | 101,925 (18.1) | 33,503 (19.0) | 86,806 (19.5) | 576,018 (18.4) |
| 65–69 years | 308,725 (17.1) | 16,334 (15.6) | 2859 (7.1) | 92,026 (16.4) | 30,398 (17.2) | 79,821 (17.9) | 530,163 (16.9) |
| 70–74 years | 309,166 (17.1) | 16,499 (15.7) | 21,297 (53.0) | 90,252 (16.1) | 31,134 (17.6) | 78,613 (17.6) | 546,961 (17.5) |
| Median (IQI) | 60 (54–67) | 59 (55–66) | 72 (56–75) | 60 (55–67) | 60 (54–67) | 61 (54–67) | 60 (54–67) |
| **Ethnicity** | | | | | | | |
| Danish descent | 1,651,658 (91.6) | 95,490 (91.1) | 36,665 (91.4) | 512,355 (91.2) | 150,194 (91.0) | 366,141 (91.2) | 2,812,503 (91.4) |
| Descendant of immigrant | 2932 (0.2) | 169 (0.2) | 65 (0.2) | 987 (0.2) | 284 (0.2) | 617 (0.2) | 5054 (0.2) |
| Western immigrant | 48,061 (2.7) | 2871 (2.7) | 1164 (2.9) | 15,877 (2.8) | 4642 (2.8) | 10,902 (2.7) | 83,517 (2.7) |
| Non-western immigrant | 99,730 (5.5) | 6334 (6.0) | 2238 (5.6) | 32,755 (5.8) | 9939 (6.0) | 23,620 (5.9) | 174,616 (5.7) |
| **Cohabitation status** | | | | | | | |
| Living alone | 567,436 (31.5) | 34,149 (32.6) | 14,487 (36.1) | 177,512 (31.6) | 51,320 (31.1) | 120,737 (30.1) | 965,641 (31.4) |
| Cohabiting/co-living | 163,186 (9.1) | 9679 (9.2) | 2758 (6.9) | 53,372 (9.5) | 15,775 (9.6) | 37,404 (9.3) | 282,174 (9.2) |
| Married/registered partner | 1,071,632 (59.5) | 61,036 (58.2) | 22,887 (57.0) | 331,090 (58.9) | 97,964 (59.4) | 243,139 (60.6) | 1,827,748 (59.4) |
| **Educational level (ISCED)** | | | | | | | |
| ISCED15 levels 1–2 | 976,876 (55.1) | 58,698 (57.0) | 18,575 (47.2) | 316,639 (57.3) | 98,365 (56.8) | 250,198 (57.2) | 1,719,351 (55.9) |
| ISCED15 levels 3–5 | 597,535 (33.7) | 33,332 (32.3) | 15,297 (38.8) | 177,249 (32.1) | 55,821 (32.2) | 139,785 (32.0) | 1,019,019 (33.1) |
| ISCED15 levels 6–8 | 198,531 (11.2) | 11,011 (10.7) | 5513 (14.0) | 58,592 (10.6) | 18,970 (11.0) | 47,211 (10.8) | 339,828 (11.0) |
| **Disposable income** | | | | | | | |
| Lowest quintile | 353,612 (19.6) | 18,722 (17.9) | 9939 (24.8) | 97,358 (17.4) | 29,966 (17.0) | 74,581 (16.8) | 584,178 (18.7) |
| Second quintile | 358,955 (19.9) | 20,579 (19.7) | 9583 (23.9) | 105,809 (18.9) | 33,736 (19.2) | 83,982 (18.9) | 612,644 (19.6) |
| Third quintile | 366,888 (20.3) | 21,023 (20.1) | 7332 (18.3) | 114,154 (20.4) | 33,347 (18.9) | 82,529 (18.6) | 625,273 (20.0) |
| Fourth quintile | 366,219 (20.3) | 21,767 (20.8) | 6614 (16.5) | 119,544 (21.3) | 36,705 (20.8) | 94,378 (21.2) | 645,227 (20.6) |
| Highest quintile | 358,608 (19.9) | 22,588 (21.6) | 6644 (16.6) | 123,676 (22.1) | 42,412 (24.1) | 108,662 (24.5) | 662,590 (21.2) |
| **Healthcare usage** | | | | | | | |
| Rare | 386,510 (21.4) | 23,944 (22.8) | 7293 (18.2) | 125,799 (22.4) | 37,849 (21.4) | 95,024 (21.3) | 676,419 (21.6) |
| Low | 338,179 (18.7) | 18,944 (18.1) | 6054 (15.1) | 103,813 (18.5) | 32,147 (18.2) | 80,134 (18.0) | 579,271 (18.5) |
| Average | 401,358 (22.2) | 23,271 (22.2) | 8569 (21.3) | 125,600 (22.3) | 39,176 (22.2) | 98,833 (22.2) | 696,807 (22.2) |
| High | 333,529 (18.5) | 19,230 (18.3) | 8343 (20.8) | 104,646 (18.6) | 33,618 (19.1) | 85,706 (19.2) | 585,072 (18.7) |
| Frequent | 345,194 (19.1) | 19,503 (18.6) | 9887 (24.6) | 102,266 (18.2) | 33,679 (19.1) | 85,849 (19.3) | 596,378 (19.0) |
| Time from invitation to participation, median (IDI) | 28 (16–72) | 25 (15–78) | 32 (16–64) | 28 (16–66) | 24 (16–63) | 25 (16–59) | 28 (16–70) |

IQI = interquartile interval.. IDI = interdecentile interval.. ISCED = International Standard Classification of Education.

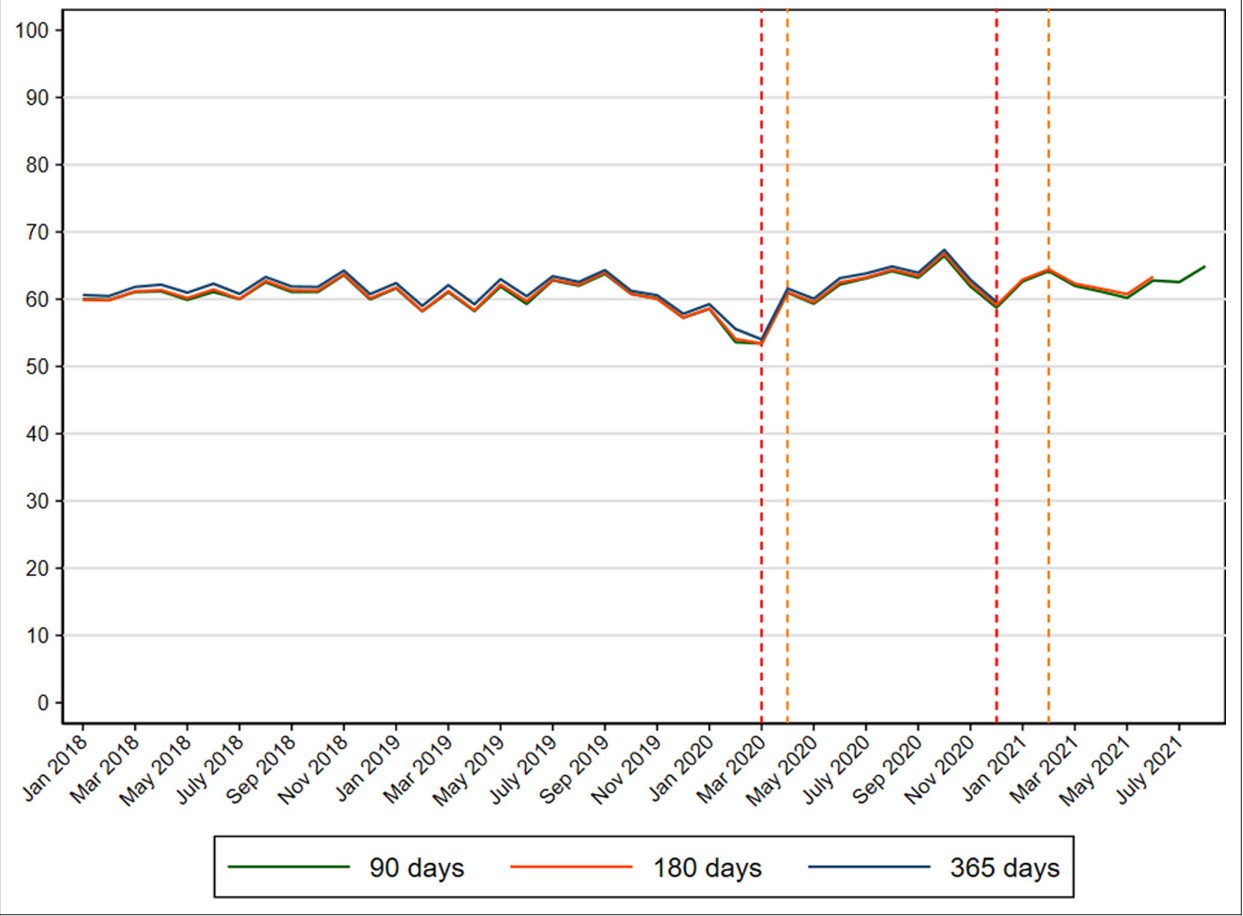

**Figure 3.** Participation in colorectal cancer screening (%) in Denmark within 90, 180 and 365 days since invitation from 2018 to 2021.

1.03–1.09), whereas the participation was similar to the previous years for the remaining part of the study period (*Supplementary file 3*).

## Compliance with colonoscopy during the COVID-19 pandemic

There were 94,373 positive FIT tests (in 92,848 individuals) during the study period. Among those 53.7% were men, the median age was 65 years old (IQI=57–70) and 93.3% were of Danish descent (*Table 3*). Before the pandemic, 89.9% had a colonoscopy performed within 60 days since a positive FIT test (*Supplementary file 4*). The results were unchanged when extending the length of follow-up time to 365 days (data not shown). A minor reduction in compliance with colonoscopy within 60 days was seen during first lockdown (*Figure 4*) reflected in a PR of 0.96 (95% CI: 0.93–0.98) (*Table 4*). The reduction corresponded to a compliance rate of 85.2% (*Supplementary file 4*). The compliance remained stable throughout the rest of the study period (*Table 4*).

Immigrants, individuals living alone and individuals with a low income had a lower compliance with colonoscopy before the pandemic (*Supplementary file 4*). During pre-lockdown and first lockdown, the compliance with colonoscopy within 60 days was lower among both 55–59 and 70–74 years old, among immigrants and among individuals with a low income compared with the previous years (*Table 4*). The results were unchanged when extending the length of follow-up time to 365 days (data not shown).

## Time to participation

Before the pandemic, the median time from invitation to participation was 28 days (IDI = 16–72) increasing to 32 days (IDI = 16–64) during first lockdown and returning to 28 days during first re-opening (*Table 1*).

**Table 2.** Prevalence ratios (PRs) and 95% confidence intervals (CIs) of participation in colorectal cancer screening within 90 days since invitation in Denmark 2018–2021*.

| | N | Pre-pandemic (1 January 2018 to 31 January 2020) N=180,477 | | Pre-lockdown (1 February 2020 to 10 March 2020) N=104,892 | | 1st lockdown (11 March 2020 to 15 April 2020) N=40,146 | | 1st re-opening (16 April 2020 to 15 December 2020) N=562,124 | | 2nd lockdown (16 December 2020 to 27 February 2021) N=176,469 | | 2nd re-opening (28 February 2021 to 30 September 2021) N=445,546 | |
|---|---|---|---|---|---|---|---|---|---|---|---|---|---|
| | N | PR | [95% CI] | PR | [95% CI] | PR | [95% CI] | PR | [95% CI] | PR | [95% CI] | PR | [95% CI] |
| Overall | 3,133,947 | 1.00 | – | 0.95 | [0.94–0.96] | 0.85 | [0.85–0.86] | 1.04 | [1.04–1.05] | 1.09 | [1.09–1.10] | 1.11 | [1.10–1.12] |
| **Sex** | | | | | | | | | | | | | |
| Men | 1,547,928 | 1.00 | – | 0.88 | [0.88–0.89] | 0.91 | [0.89–0.92] | 1.03 | [1.02–1.03] | 1.04 | [1.04–1.05] | 1.07 | [1.07–1.08] |
| Women | 1,586,019 | 1.00 | – | 0.92 | [0.92–0.93] | 0.84 | [0.83–0.85] | 1.03 | [1.02–1.03] | 1.03 | [1.03–1.04] | 1.06 | [1.06–1.07] |
| **Age at invitation** | | | | | | | | | | | | | |
| 50–54 years | 816,619 | 1.00 | – | 0.89 | [0.87–0.91] | 1.14 | [1.12–1.17] | 1.00 | [0.99–1.01] | 1.04 | [1.03–1.06] | 1.16 | [1.14–1.18] |
| 55–59 years | 664,186 | 1.00 | – | 0.93 | [0.91–0.94] | 1.09 | [1.06–1.11] | 1.11 | [1.10–1.12] | 1.24 | [1.22–1.25] | 1.18 | [1.16–1.19] |
| 60–64 years | 576,018 | 1.00 | – | 0.94 | [0.93–0.96] | 1.03 | [1.00–1.06] | 1.03 | [1.02–1.04] | 1.08 | [1.07–1.10] | 1.09 | [1.07–1.10] |
| 65–69 years | 530,163 | 1.00 | – | 0.96 | [0.95–0.97] | 0.99 | [0.96–1.02] | 1.02 | [1.01–1.03] | 1.07 | [1.06–1.09] | 1.07 | [1.06–1.08] |
| 70–74 years | 546,961 | 1.00 | – | 1.01 | [1.00–1.03] | 0.77 | [0.76–0.79] | 1.04 | [1.03–1.05] | 1.09 | [1.07–1.10] | 1.10 | [1.08–1.11] |
| **Ethnicity** | | | | | | | | | | | | | |
| Danish descent | 2,812,503 | 1.00 | – | 0.95 | [0.94–0.95] | 0.85 | [0.84–0.86] | 1.04 | [1.04–1.05] | 1.09 | [1.08–1.10] | 1.11 | [1.10–1.12] |
| Descendant of immigrant | 5054 | 1.00 | – | 0.95 | [0.79–1.14] | 0.83 | [0.62–1.11] | 1.04 | [0.93–1.17] | 1.08 | [0.91–1.28] | 1.12 | [0.94–1.32] |
| Western Immigrant | 83,517 | 1.00 | – | 0.95 | [0.91–1.00] | 0.88 | [0.82–0.94] | 1.08 | [1.05–1.11] | 1.16 | [1.11–1.21] | 1.20 | [1.15–1.25] |
| Non-western immigrant | 174,616 | 1.00 | – | 0.97 | [0.94–1.00] | 1.00 | [0.95–1.05] | 1.11 | [1.09–1.13] | 1.18 | [1.15–1.22] | 1.24 | [1.20–1.28] |
| **Cohabitation status** | | | | | | | | | | | | | |
| Living alone | 965,641 | 1.00 | – | 0.94 | [0.93–0.95] | 0.85 | [0.83–0.87] | 1.06 | [1.05–1.07] | 1.12 | [1.11–1.14] | 1.16 | [1.14–1.17] |
| Cohabiting/co-living | 282,174 | 1.00 | – | 0.95 | [0.92–0.97] | 0.97 | [0.93–1.00] | 1.05 | [1.04–1.07] | 1.12 | [1.09–1.14] | 1.14 | [1.12–1.17] |
| Married/registered partner | 1,827,748 | 1.00 | – | 0.95 | [0.95–0.96] | 0.86 | [0.85–0.87] | 1.04 | [1.03–1.04] | 1.08 | [1.07–1.09] | 1.09 | [1.09–1.10] |
| **Educational level (ISCED)** | | | | | | | | | | | | | |
| ISCED15 levels 1–2 | 1,719,351 | 1.00 | – | 0.95 | [0.94–0.95] | 0.91 | [0.90–0.92] | 1.05 | [1.04–1.05] | 1.09 | [1.08–1.10] | 1.12 | [1.11–1.13] |
| ISCED15 levels 3–5 | 1,019,019 | 1.00 | – | 0.96 | [0.94–0.97] | 0.80 | [0.79–0.82] | 1.04 | [1.03–1.04] | 1.09 | [1.08–1.10] | 1.10 | [1.09–1.11] |
| ISCED15 levels 6–8 | 339,828 | 1.00 | – | 0.95 | [0.94–0.97] | 0.83 | [0.81–0.85] | 1.05 | [1.04–1.06] | 1.11 | [1.09–1.13] | 1.12 | [1.10–1.13] |
| **Disposable income** | | | | | | | | | | | | | |
| Lowest quintile | 584,178 | 1.00 | – | 0.95 | [0.94–0.97] | 0.73 | [0.71–0.75] | 1.04 | [1.03–1.05] | 1.08 | [1.06–1.10] | 1.11 | [1.09–1.12] |
| Second quintile | 612,644 | 1.00 | – | 0.96 | [0.95–0.98] | 0.78 | [0.76–0.79] | 1.05 | [1.04–1.06] | 1.10 | [1.08–1.11] | 1.13 | [1.11–1.14] |
| Third quintile | 625,273 | 1.00 | – | 0.95 | [0.94–0.97] | 0.86 | [0.84–0.88] | 1.05 | [1.04–1.06] | 1.08 | [1.07–1.10] | 1.11 | [1.10–1.12] |
| Fourth quintile | 645,227 | 1.00 | – | 0.95 | [0.94–0.96] | 0.94 | [0.92–0.96] | 1.04 | [1.04–1.05] | 1.08 | [1.07–1.10] | 1.10 | [1.09–1.11] |
| Highest quintile | 662,590 | 1.00 | – | 0.93 | [0.92–0.94] | 0.98 | [0.96–1.00] | 1.04 | [1.03–1.05] | 1.10 | [1.09–1.11] | 1.10 | [1.09–1.11] |
| **Healthcare usage** | | | | | | | | | | | | | |
| Rare | 676,419 | 1.00 | – | 0.92 | [0.90–0.93] | 0.93 | [0.90–0.95] | 1.04 | [1.03–1.05] | 1.10 | [1.08–1.12] | 1.11 | [1.09–1.12] |
| Low | 579,271 | 1.00 | – | 0.95 | [0.94–0.97] | 0.90 | [0.88–0.92] | 1.05 | [1.04–1.06] | 1.10 | [1.08–1.11] | 1.11 | [1.10–1.13] |
| Average | 696,807 | 1.00 | – | 0.95 | [0.94–0.97] | 0.86 | [0.85–0.88] | 1.05 | [1.04–1.06] | 1.09 | [1.08–1.10] | 1.11 | [1.10–1.12] |
| High | 585,072 | 1.00 | – | 0.97 | [0.95–0.98] | 0.84 | [0.82–0.86] | 1.05 | [1.04–1.05] | 1.09 | [1.08–1.11] | 1.12 | [1.10–1.13] |
| Frequent | 596,378 | 1.00 | – | 0.97 | [0.95–0.98] | 0.81 | [0.79–0.83] | 1.04 | [1.03–1.05] | 1.09 | [1.08–1.10] | 1.11 | [1.10–1.12] |

*Adjusted for year, month and age at invitation; PR = prevalence ratio; CI = confidence interval; ISCED = International Standard Classification of Education.

**Table 3.** Baseline characteristics of people with a positive FIT test from colorectal cancer screening in Denmark from 2018 to 2021.

| | Pre-pandemic (1 January 2018 to 31 January 2020) | Pre-lockdown (1 February 2020 to 10 March 2020) | 1st lockdown (11 March 2020 to 15 April 2020) | 1st re-opening (16 April 2020 to 15 December 2020) | 2nd lockdown (16 December 2020 to 27 Febraury 2021) | 2nd re-opening (28 February 2021 to 30 September 2021) | Total |
|---|---|---|---|---|---|---|---|
| | N (%) | N (%) | N (%) | N (%) | N (%) | N (%) | N (%) |
| Total | 54,886 (58.2) | 2942 (3.1) | 1319 (1.4) | 16,628 (17.6) | 5383 (5.7) | 13,215 (14.0) | 94,373 (100.0) |
| **Sex** | | | | | | | |
| Men | 29,880 (54.4) | 1532 (52.1) | 704 (53.4) | 8796 (52.9) | 2835 (52.7) | 6918 (52.3) | 50,665 (53.7) |
| Women | 25,006 (45.6) | 1410 (47.9) | 615 (46.6) | 7832 (47.1) | 2548 (47.3) | 6297 (47.7) | 43,708 (46.3) |
| **Age at invitation** | | | | | | | |
| 50–54 years | 9481 (17.3) | 291 (9.9) | 185 (14.0) | 2615 (15.7) | 987 (18.3) | 2876 (21.8) | 16,435 (17.4) |
| 55–59 years | 8606 (15.7) | 838 (28.5) | 157 (11.9) | 3536 (21.3) | 813 (15.1) | 1479 (11.2) | 15,429 (16.3) |
| 60–64 years | 10,343 (18.8) | 524 (17.8) | 78 (5.9) | 2944 (17.7) | 1000 (18.6) | 2433 (18.4) | 17,322 (18.4) |
| 65–69 years | 12,126 (22.1) | 559 (19.0) | 111 (8.4) | 3526 (21.2) | 1165 (21.6) | 2888 (21.9) | 20,375 (21.6) |
| 70–74 years | 14,330 (26.1) | 730 (24.8) | 788 (59.7) | 4007 (24.1) | 1418 (26.3) | 3539 (26.8) | 24,812 (26.3) |
| Median (IQI) | 64 (57–70) | 63 (56–69) | 74 (59–75) | 63 (56–69) | 64 (57–70) | 64 (57–70) | 64 (57–70) |
| **Ethnicity** | | | | | | | |
| Danish descent | 51,287 (93.6) | 2735 (93.0) | 1221 (92.6) | 15,484 (93.1) | 4716 (92.4) | 11,344 (92.5) | 86,787 (93.3) |
| Western immigrant | 1306 (2.4) | 66 (2.2) | 37 (2.8) | 388 (2.3) | 130 (2.5) | 302 (2.5) | 2229 (2.4) |
| Non-western immigrant | 2205 (4.0) | 140 (4.8) | 61 (4.6) | 756 (4.5) | 257 (5.0) | 620 (5.1) | 4039 (4.3) |
| **Cohabitation status** | | | | | | | |
| Living alone | 16,451 (30.0) | 939 (31.9) | 463 (35.1) | 5046 (30.3) | 1578 (30.9) | 3575 (29.1) | 28,052 (30.1) |
| Cohabiting/co-living | 4273 (7.8) | 231 (7.9) | 88 (6.7) | 1377 (8.3) | 431 (8.4) | 1037 (8.5) | 7437 (8.0) |
| Married/registered partner | 34,074 (62.2) | 1771 (60.2) | 768 (58.2) | 10,205 (61.4) | 3094 (60.6) | 7654 (62.4) | 57,566 (61.9) |
| **Educational level (ISCED)** | | | | | | | |
| ISCED15 levels 1–2 | 27,676 (51.3) | 1577 (54.6) | 605 (46.6) | 8988 (54.9) | 2823 (53.6) | 7064 (54.4) | 48,733 (52.5) |
| ISCED15 levels 3–5 | 20,618 (38.2) | 1048 (36.3) | 521 (40.1) | 5741 (35.1) | 1907 (36.2) | 4682 (36.1) | 34,517 (37.2) |
| ISCED15 levels 6–8 | 5666 (10.5) | 264 (9.1) | 172 (13.3) | 1632 (10.0) | 540 (10.2) | 1231 (9.5) | 9505 (10.2) |
| **Disposable income** | | | | | | | |
| Lowest quintile | 11,851 (21.6) | 556 (18.9) | 321 (24.3) | 3124 (18.8) | 962 (17.9) | 2481 (18.8) | 19,295 (20.5) |
| Second quintile | 12,530 (22.8) | 690 (23.5) | 351 (26.6) | 3555 (21.4) | 1211 (22.5) | 2954 (22.4) | 21,291 (22.6) |
| Third quintile | 11,077 (20.2) | 608 (20.7) | 234 (17.7) | 3439 (20.7) | 1042 (19.4) | 2479 (18.8) | 18,879 (20.0) |

*Table 3 continued on next page*

*Table 3 continued*

| | Pre-pandemic (1 January 2018 to 31 January 2020) | Pre-lockdown (1 February 2020 to 10 March 2020) | 1st lockdown (11 March 2020 to 15 April 2020) | 1st re-opening (16 April 2020 to 15 December 2020) | 2nd lockdown (16 December 2020 to 27 Febraury 2021) | 2nd re-opening (28 February 2021 to 30 September 2021) | Total |
|---|---|---|---|---|---|---|---|
| | N (%) | N (%) | N (%) | N (%) | N (%) | N (%) | N (%) |
| Fourth quintile | 10,142 (18.5) | 569 (19.4) | 210 (15.9) | 3321 (20.0) | 1046 (19.5) | 2615 (19.9) | 17,903 (19.0) |
| Highest quintile | 9244 (16.9) | 517 (17.6) | 203 (15.4) | 3165 (19.1) | 1114 (20.7) | 2641 (20.1) | 16,884 (17.9) |
| Healthcare usage | | | | | | | |
| Rare | 8431 (15.4) | 409 (13.9) | 161 (12.2) | 2576 (15.5) | 770 (14.3) | 1903 (14.4) | 14,250 (15.1) |
| Low | 8450 (15.4) | 440 (15.0) | 171 (13.0) | 2628 (15.8) | 833 (15.5) | 1957 (14.8) | 14,479 (15.3) |
| Average | 11,964 (21.8) | 608 (20.7) | 260 (19.7) | 3704 (22.3) | 1205 (22.4) | 2823 (21.4) | 20,564 (21.8) |
| High | 11,300 (20.6) | 673 (22.9) | 269 (20.4) | 3502 (21.1) | 1126 (20.9) | 2921 (22.1) | 19,791 (21.0) |
| Frequent | 14,741 (26.9) | 812 (27.6) | 458 (34.7) | 4218 (25.4) | 1449 (26.9) | 3611 (27.3) | 25,289 (26.8) |
| Time from positive FIT to colonoscopy, median (IDI) | 13 (8–29) | 13 (7–31) | 13 (8–31) | 13 (8–29) | 13 (8–28) | 13 (8–29) | 13 (8–29) |

IQI = interquartile interval.. IDI = interdecentile interval.. ISCED = International Standard Classification of Education.

## Sensitivity analysis

When conducting a sensitivity analysis accounting for the reduction in the number of invitations in weeks 11–14 2020, the estimates were almost identical (data not shown).

## Discussion
### Main findings

In this nationwide population-based study comprising more than 3.1 million invitations (in 1,928,725 individuals), we found that 60.7% participated in colorectal cancer screening within 90 days before the pandemic. A minor reduction in participation was observed at the start of the pandemic corresponding to a participation rate of 54.9% during pre-lockdown and 53.0% during first lockdown. From the first re-opening of the society and onwards, a relative 5–10% increased participation in screening was seen corresponding to a participation rate of up to 64.9%. The largest relative increase in participation was observed among 55–59 years old and among immigrants. Among 94,373 positive FIT tests (in 92,848 individuals), we saw that the compliance with colonoscopy within 60 days was 89.9% before the pandemic. A slight reduction was observed during first lockdown, where after it resumed to normal levels.

### Comparison with previous studies

In most of the world, the colorectal cancer screening programmes were paused at the start of the pandemic (*Morris et al., 2021*; *Dinmohamed et al., 2020*; *Kortlever et al., 2021*; *Vives et al., 2022*; *Walker et al., 2021*), while Denmark was one of the only countries to have the programme running throughout the pandemic. The situation in Denmark is therefore unique and resembles 'a natural experiment' illustrating what happens if a screening programme based on faecal samples obtained at home is kept open during a pandemic.

In Catalonia in Spain, the colorectal cancer screening programme was paused for the first 7 months of the pandemic leading to a reduced participation in stool-based screening (5.1% reduction) and an increased proportion of advanced-stage cancers (stages III–IV) (13% increase) (*Vives et al., 2022*).

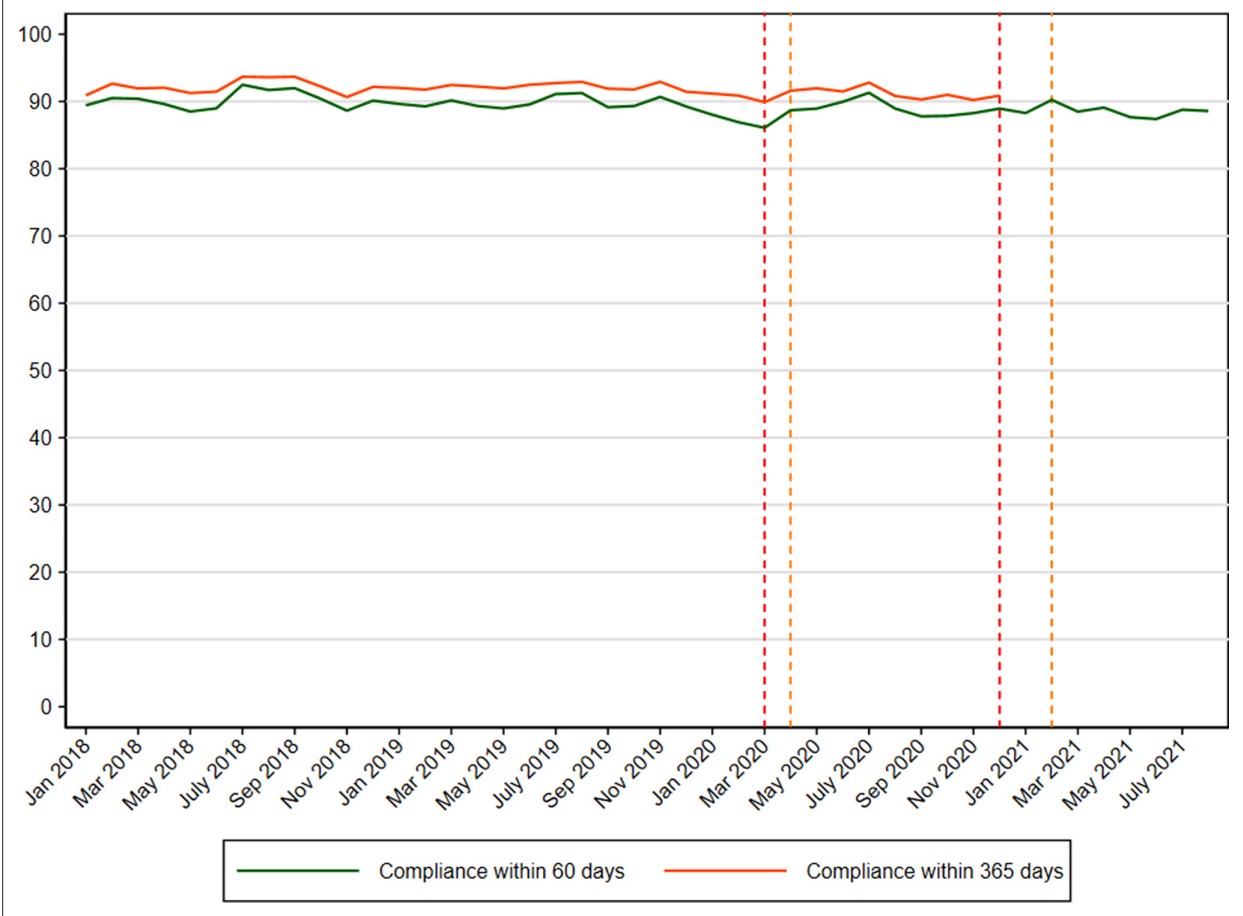

**Figure 4.** Compliance with colonoscopy (%) within 60 and 365 days since a positive FIT test from colorectal cancer screening in Denmark from 2018 to 2021.

The screening programme in Catalonia in Spain (***Vives et al., 2022***) requires the invitees to collect a FIT test kit at a pharmacy, which might be an additional barrier to participation in particular during a pandemic. The programme in Denmark is on the contrary based on a home-based test mailed directly to the invitees. Also in England, the colorectal cancer screening programme based on faecal samples was paused at the start of the pandemic, which led to large reductions in the number of people referred for suspected cancer (63% relative reduction), diagnosed (22% relative reduction) and receiving surgery for colorectal cancer (31% relative reduction) in April 2020 (***Morris et al., 2021***). In the Netherlands, the colorectal cancer screening programme was temporarily halted at the start of the pandemic resulting in a lower participation in FIT screening (5% and 7% reduction in February and March 2020, respectively) and fewer colorectal cancers diagnosed (***Dinmohamed et al., 2020***). In Canada, the FIT-based screening programme was suspended at the start of the pandemic resulting in a large reduction in the colorectal cancer faecal test volume (90% reduction in May 2020) (***Walker et al., 2021***). We found a minor reduction in participation at the start of the pandemic corresponding to a participation rate of 54.9% during pre-lockdown and 53.0% during first lockdown indicating that a few people do not participate at the very early phase of a pandemic. These results did not differ when extending the length of follow-up time.

The study from Catalonia in Spain ***Vives et al., 2022*** found a reduction in participation in subsequent colonoscopy (8.9% reduction) as a result of the screening programme being closed during the first 7 months of the pandemic. The study from England also showed a marked reduction in the number of colonoscopies (92% relative reduction) (***Morris et al., 2021***) as a result of the screening programme being paused at the start of the pandemic. The study from the Netherlands showed a reduced participation in follow-up colonoscopy in the months before and during the suspension of the FIT screening programme (***Kortlever et al., 2021***). The suspension of the colorectal cancer

**Table 4.** Prevalence ratios (PRs) and 95% confidence intervals (CIs) of compliance with colonoscopy within 60 days since a positive FIT test from colorectal cancer screening in Denmark from 2018 to 2021*.

| | | Pre-pandemic (1 January 2018 to 31 January 2020) N=54,982 | | Pre-lockdown (1 February 2020 to 10 March 2020) N=2944 | | 1st lockdown (11 March 2020 to 15 April 2020) N=1319 | | 1st re-opening (16 April 2020 to 15 December 2020) N=16,628 | | 2nd lockdown (16 December 2020 to 27 February 2021) N=5390 | | 2nd re-opening (28 February 2021 to 30 September 2021) N=13,222 | |
|---|---|---|---|---|---|---|---|---|---|---|---|---|---|
| | N | PR | [95% CI] | PR | [95% CI] | PR | [95% CI] | PR | [95% CI] | PR | [95% CI] | PR | [95% CI] |
| Overall | 94,373 | 1.00 | – | 0.98 | [0.96–1.00] | 0.96 | [0.93–0.98] | 0.99 | [0.98–1.00] | 1.01 | [1.00–1.03] | 0.99 | [0.98–1.01] |
| **Sex** | | | | | | | | | | | | | |
| Men | 50,665 | 1.00 | – | 0.97 | [0.95–0.99] | 0.94 | [0.91–0.97] | 0.99 | [0.98–0.99] | 0.99 | [0.97–1.00] | 0.97 | [0.96–0.98] |
| Women | 43,708 | 1.00 | – | 0.98 | [0.96–1.00] | 0.96 | [0.93–0.99] | 0.99 | [0.98–1.00] | 1.00 | [0.99–1.02] | 0.99 | [0.98–1.00] |
| **Age at invitation** | | | | | | | | | | | | | |
| 50–54 years | 16,435 | 1.00 | – | 1.04 | [1.00–1.08] | 0.98 | [0.93–1.04] | 1.00 | [0.98–1.03] | 1.01 | [0.97–1.05] | 0.98 | [0.94–1.01] |
| 55–59 years | 15,429 | 1.00 | – | 0.97 | [0.94–1.01] | 0.94 | [0.87–1.01] | 1.00 | [0.98–1.02] | 1.02 | [0.98–1.05] | 1.02 | [0.98–1.06] |
| 60–64 years | 17,322 | 1.00 | – | 1.00 | [0.97–1.04] | 1.04 | [0.98–1.11] | 0.99 | [0.97–1.01] | 1.00 | [0.97–1.04] | 0.98 | [0.94–1.01] |
| 65–69 years | 20,375 | 1.00 | – | 0.99 | [0.96–1.03] | 1.04 | [0.98–1.10] | 0.99 | [0.97–1.01] | 1.03 | [1.00–1.06] | 1.00 | [0.97–1.03] |
| 70–74 years | 24,812 | 2.00 | – | 0.94 | [0.91–0.98] | 0.94 | [0.90–0.97] | 0.98 | [0.96–1.00] | 1.01 | [0.98–1.04] | 0.99 | [0.96–1.02] |
| **Ethnicity** | | | | | | | | | | | | | |
| Danish descent | 86,787 | 1.00 | – | 0.99 | [0.97–1.00] | 0.96 | [0.94–0.99] | 1.00 | [0.99–1.01] | 1.02 | [1.00–1.03] | 1.00 | [0.98–1.01] |
| Western immigrant | 2229 | 1.00 | – | 0.87 | [0.75–1.02] | 0.84 | [0.68–1.04] | 0.97 | [0.90–1.05] | 0.99 | [0.87–1.13] | 0.99 | [0.89–1.11] |
| Non-western immigrant | 4039 | 1.00 | – | 0.91 | [0.82–1.01] | 0.90 | [0.78–1.05] | 0.98 | [0.92–1.04] | 1.00 | [0.92–1.10] | 0.95 | [0.87–1.04] |
| **Cohabitation status** | | | | | | | | | | | | | |
| Living alone | 28,052 | 1.00 | – | 0.97 | [0.93–1.00] | 0.95 | [0.90–1.00] | 1.00 | [0.97–1.02] | 1.04 | [1.01–1.08] | 1.04 | [1.01–1.08] |
| Cohabiting/co-living | 7437 | 1.00 | – | 1.02 | [0.96–1.08] | 1.01 | [0.93–1.09] | 1.01 | [0.98–1.05] | 1.03 | [0.98–1.08] | 1.03 | [0.98–1.08] |
| Married/registered partner | 57,566 | 1.00 | – | 0.99 | [0.97–1.00] | 0.96 | [0.93–0.99] | 0.99 | [0.98–1.00] | 1.00 | [0.98–1.02] | 1.00 | [0.98–1.02] |
| **Educational level (ISCED)** | | | | | | | | | | | | | |
| ISCED15 levels 1–2 | 48,733 | 1.00 | – | 1.00 | [0.98–1.02] | 0.94 | [0.91–0.98] | 0.99 | [0.97–1.00] | 1.01 | [0.99–1.03] | 0.98 | [0.96–1.00] |
| ISCED15 levels 3–5 | 34517 | 1.00 | – | 0.97 | [0.95–1.00] | 0.97 | [0.94–1.01] | 1.00 | [0.99–1.02] | 1.01 | [0.99–1.04] | 1.01 | [0.98–1.03] |
| ISCED15 levels 6–8 | 9505 | 1.00 | – | 0.94 | [0.89–1.00] | 0.97 | [0.91–1.03] | 0.99 | [0.96–1.02] | 1.02 | [0.98–1.07] | 0.99 | [0.94–1.03] |
| **Disposable income** | | | | | | | | | | | | | |
| Lowest quintile | 19,295 | 1.00 | – | 0.96 | [0.92–1.00] | 0.91 | [0.86–0.96] | 0.99 | [0.96–1.01] | 1.02 | [0.98–1.06] | 0.99 | [0.95–1.02] |
| Second quintile | 21,291 | 1.00 | – | 0.98 | [0.94–1.02] | 0.92 | [0.87–0.98] | 1.01 | [0.99–1.04] | 1.04 | [1.01–1.08] | 1.02 | [0.99–1.06] |
| Third quintile | 18,879 | 1.00 | – | 0.99 | [0.95–1.02] | 0.98 | [0.93–1.03] | 0.98 | [0.96–1.00] | 0.99 | [0.96–1.03] | 0.99 | [0.96–1.02] |
| Fourth quintile | 17,903 | 1.00 | – | 0.97 | [0.94–1.00] | 0.98 | [0.93–1.02] | 0.99 | [0.97–1.01] | 1.00 | [0.97–1.03] | 0.98 | [0.95–1.01] |
| Highest quintile | 16,884 | 1.00 | – | 1.01 | [0.98–1.05] | 1.02 | [0.97–1.06] | 0.99 | [0.97–1.01] | 1.00 | [0.97–1.03] | 0.98 | [0.95–1.00] |
| **Healthcare usage** | | | | | | | | | | | | | |
| Rare | 14,250 | 1.00 | – | 0.98 | [0.94–1.02] | 0.98 | [0.92–1.04] | 1.01 | [0.99–1.04] | 1.03 | [0.99–1.07] | 0.99 | [0.96–1.03] |
| Low | 14,479 | 1.00 | – | 0.97 | [0.93–1.01] | 0.97 | [0.92–1.03] | 0.99 | [0.97–1.02] | 0.99 | [0.95–1.02] | 0.98 | [0.95–1.01] |
| Average | 20,564 | 1.00 | – | 1.01 | [0.98–1.04] | 0.94 | [0.89–0.99] | 1.00 | [0.98–1.02] | 1.02 | [0.99–1.05] | 0.99 | [0.96–1.02] |

*Table 4 continued on next page*

*Table 4 continued*

| | N | Pre-pandemic (1 January 2018 to 31 January 2020) N=54,982 | | Pre-lockdown (1 February 2020 to 10 March 2020) N=2944 | | 1st lockdown (11 March 2020 to 15 April 2020) N=1319 | | 1st re-opening (16 April 2020 to 15 December 2020) N=16,628 | | 2nd lockdown (16 December 2020 to 27 February 2021) N=5390 | | 2nd re-opening (28 February 2021 to 30 September 2021) N=13,222 | |
|---|---|---|---|---|---|---|---|---|---|---|---|---|---|
| | | PR | [95% CI] | PR | [95% CI] | PR | [95% CI] | PR | [95% CI] | PR | [95% CI] | PR | [95% CI] |
| High | 19,791 | 1.00 | – | 0.99 | [0.95–1.02] | 0.97 | [0.92–1.01] | 0.98 | [0.96–1.01] | 1.01 | [0.97–1.04] | 1.01 | [0.97–1.04] |
| Frequent | 25,289 | 1.00 | – | 0.96 | [0.93–1.00] | 0.95 | [0.90–0.99] | 0.98 | [0.96–1.01] | 1.01 | [0.98–1.04] | 0.98 | [0.95–1.01] |

*Adjusted for year, month and age at invitation; PR = prevalence ratio; CI = confidence interval; ISCED = International Standard Classification of Education.

screening programme in Canada led to a 99% reduction of colonoscopies performed in April 2020 (*Walker et al., 2021*). We found a 4% reduction in compliance with colonoscopy within 60 days during first lockdown despite the programme being open. Fortunately, the compliance with colonoscopy resumed to the same level as before the pandemic from first re-opening and onwards. Congruently, a qualitative study from the United Kingdom *Wilson et al., 2021* found that interview participants were concerned about visiting healthcare settings at the start of the pandemic, which could explain the slightly lower compliance with colonoscopy.

The potential downstream effect of a reduced or increased participation in colorectal cancer screening during the pandemic on subsequent colorectal cancer diagnoses and deaths are important aspects to take into account. A Danish study by Skovlund et al. found a 24% reduction in the number of colon cancers diagnosed from April to June 2020 in Denmark (*Skovlund et al., 2022*) which is most likely an overestimation due to a delayed registration of cancers at the time of the study by Skovlund et al. In a more recent study by Weinberger et al. using data from the Danish Colorectal Cancer Database (*Ingeholm et al., 2016*), we have shown a drop in the number of colorectal cancers detected during first lockdown; however, only a minor reduction (7%) of colorectal cancers in 2020 as compared to the previous years (Weinberger et al., under review). This reduction may be caused either by a reduced participation in colorectal cancer screening, a change in health-seeking behaviour with fever referred to colonoscopy after symptoms or a reduction of coincidental detections of colorectal cancers as a result of elective surveillance colonoscopy being cancelled or postponed. Thus in the study by Weinberger et al. (under review), we found a reduction in the proportion of colorectal cancers detected via screening (16.9% vs. 21.8%; PR = 0.79; 95%CI: 0.73–0.86) during the pandemic. Furthermore, we observed a slight increase in the proportion of stage I tumours (25.0% vs. 23.4%; PR = 1.07; 95%CI: 1.00–1.15); however, the risk of death within 90days since operation was unchanged (3.9% vs. 3.6%; PR = 1.02; 95%CI: 0.84–1.23) during the pandemic compared with the previous years (Weinberger et al., under review). These results are thus based on a setting where FIT-based colorectal cancer screening and early detection of cancer in general continued throughout the pandemic. Modelling studies from other countries have shown that short-term disruptions to colorectal cancer screening during the pandemic would have a marked impact on colorectal cancer incidence and deaths from 2020 to 2050 because of missed screening (*de Jonge et al., 2021*; *Mandrik et al., 2022*). Where possible, it is thus important to keep the FIT-based colorectal cancer screening programme running throughout a pandemic or a similar health crisis. Furthermore, catch-up screening should be provided in settings where the colorectal cancer screening programme was disrupted to mitigate the impact on subsequent colorectal cancer deaths (*de Jonge et al., 2021*).

## Socio-economic differences

In line with previous studies (*Larsen et al., 2017–Pallesen et al., 2021*), we found that the participation in colorectal cancer screening in general was lowest among the youngest age group, among men, among immigrants, among individuals living alone or cohabiting, among individuals with a low educational level, a low income and among individuals who rarely use the healthcare system.

We found an overall 5–10% increased participation in screening from first re-opening and onwards. The largest increases in participation was observed among individuals aged 55–59 years old and

among immigrants. The reason for the increased participation among individuals aged 55–59 years old is unknown. A qualitative study from Denmark found that immigrants generally have a mistrust in the Danish healthcare system and, for example, prefer a second opinion in their native country (*Tatari et al., 2020*). However, during the pandemic immigrants may not have been able to travel to their home country and may therefore have opted to participate in screening in Denmark instead of in their home-country.

### Strengths and limitations

We used high-quality population-based data covering the Danish population invited to participate in colorectal cancer screening, which is a major strength of the study. The completeness of the Danish registries is high (*Thygesen et al., 2011*), which also confers to the Danish Colorectal Cancer Screening Database (*Thomsen et al., 2017*).

Limitations of the study should also be acknowledged. We did not have data on underlying disease, which may affect individuals' participation in colorectal cancer screening and compliance with colonoscopy. We did; however, include age which is strongly associated with the level of comorbidity and we thereby reduce the theoretical impact of comorbidity on the results. Additionally, we did not have data on COVID-19 vaccination status, which may affect colorectal cancer screening participation and compliance with colonoscopy.

### Implications

The initial colorectal cancer screening test is a home-based self-collected sample and does thereby not require contact with the healthcare system—or use of, for example, public transport to reach a healthcare facility. In theory, this screening modality should be little affected by the pandemic; nonetheless, a slight reduction in participation was observed at the start of the pandemic perhaps because people counterbalance the importance of screening participation amidst a pandemic as seen in other screening programmes (*Kirkegaard et al., 2021*). It is therefore important to ensure that the health communication is clear and that people are made aware that the colorectal cancer screening programme is open and that people have the possibility to participate in both the initial screening test and in the subsequent colonoscopy.

We found an overall 5–10% increased participation in colorectal cancer screening from first re-opening and onwards. We did not find an increased participation in the other cancer screening programmes (breast and cervical cancer) in Denmark (*Olesen et al., 2023*; *Olesen et al., 2022*) most likely because those programmes require contact with the healthcare system. A home-based self-collected screening sample thus appear to work well during a pandemic, which could be used in other programmes, for example, cervical cancer screening.

The overall compliance with colonoscopy was largely unaffected by the pandemic, which gives reassurance that individuals in need of follow-up because of a positive FIT test do attend colonoscopy amidst a pandemic. Nonetheless, we found that the compliance with colonoscopy was reduced among 55–59 years , 70–74 years old, among immigrants and among persons with a low income indicating that some groups are affected adversely by the pandemic. Worryingly, some cancers may have gone undetected or diagnosed at a later stage in these groups of individuals.

### Conclusion

In this nationwide study, we found that the participation in the Danish FIT-based colorectal cancer screening programme and subsequent compliance to colonoscopy after a positive FIT test was only slightly affected by the COVID-19 pandemic.

## Acknowledgements

The study was funded by the Danish Cancer Society Scientific Committee (Grant number R321-A17417) and the Danish regions.

## Additional information

### Funding

| Funder | Grant reference number | Author |
|---|---|---|
| The Danish Cancer Society | R321-A17417 | Tina Bech Olesen<br>Henrik Møller |
| The Danish regions | | Tina Bech Olesen<br>Henry Jensen |

The funders had no role in study design, data collection and interpretation, or the decision to submit the work for publication.

### Author contributions

Tina Bech Olesen, Conceptualization, Data curation, Funding acquisition, Methodology, Writing – original draft, Writing – review and editing; Henry Jensen, Conceptualization, Data curation, Formal analysis, Methodology, Writing – review and editing; Henrik Møller, Jens Winther Jensen, Morten Rasmussen, Conceptualization, Funding acquisition, Writing – review and editing; Berit Andersen, Conceptualization, Supervision, Funding acquisition, Methodology, Writing – review and editing

### Author ORCIDs

Tina Bech Olesen http://orcid.org/0000-0002-6295-7399
Henry Jensen http://orcid.org/0000-0003-4040-7334

### Ethics

Human subjects: Ethical considerations The study is registered at the Central Denmark Region's register of research projects (journal number 1-16-02-381-20). Patient consent is not required by Danish law for register-based studies.

### Decision letter and Author response

Decision letter https://doi.org/10.7554/eLife.81808.sa1
Author response https://doi.org/10.7554/eLife.81808.sa2

## Additional files

### Supplementary files

• Supplementary file 1. Participation in colorectal cancer screening (%) in Denmark within 90 days since invitation from 2018–2021.

• Supplementary file 2. Baseline characteristics of first-time invitees invited to participate in colorectal cancer screening in Denmark 2018–2021.

• Supplementary file 3. Prevalence ratios (PRs) and 95% confidence intervals (CI) of participation in colorectal cancer screening within 90 days since invitation in Denmark among first-time invitees 2018–2021.

• Supplementary file 4. Compliance with colonoscopy (%) in Denmark within 60 days since a positive FIT test from 2018 to 2021.

• MDAR checklist

### Data availability

In order to comply with the Danish regulations on data privacy, the datasets generated and analysed during this project are not publicly available as the data are stored and maintained electronically at Statistics Denmark, where it only can be accessed by pre-approved researchers using a secure VPN remote access. Furthermore, no data at a personal level nor data not exclusively necessary for publication are allowed to be extracted from the secure data environment at Statistics Denmark. Access to the data can however be granted by the authors of the present study upon a reasonable scientific proposal within the boundaries of the present project and for scientific purposes only.

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
