## [Editor Report]

The authors convincingly demonstrate that, in the absence of any shutdowns, the Danish colorectal cancer screening program experienced only minor decreases in program participation during the COVID-19 pandemic period. This likely ensured ongoing program effectiveness in detecting early colorectal cancers and precancerous polyps. The evidence is solid and may serve as guidance for other countries when facing similar public health threats in the future.

---

## [Decision Letter]

**Decision letter after peer review:**

Thank you for submitting your article "Nation-wide participation in FIT-based colorectal cancer screening in Denmark during the COVID-19 pandemic: An observational study" for consideration by *eLife*. Your article has been reviewed by 2 peer reviewers, and the evaluation has been overseen by myself in a dual role of Reviewing Editor and Senior Editor. The following individual involved in review of your submission has agreed to reveal their identity: Paolo Giorgi Rossi (Reviewer #2).

Essential revisions:

As is customary in *eLife*, the reviewers have discussed their critiques with one another and with the Reviewing and Senior Editors. The decision was reached by consensus. What follows below is an edited compilation of the essential and ancillary points provided by reviewers in their critiques and in their interaction post-review. Please submit a revised version that addresses these concerns directly. Although we expect that you will address these comments in your response letter, we also need to see the corresponding revision clearly marked in the text of the manuscript. Some of the reviewers' comments may seem to be simple queries or challenges that do not prompt revisions to the text. Please keep in mind, however, that readers may have the same perspective as the reviewers. Therefore, it is essential that you amend or expand the text to clarify the narrative accordingly.

*Reviewer #1 (Recommendations for the authors):*

It would be useful to include a reference for the assertion that "most" programmes were paused – these references were included in the discussion but not in the introduction.

The language around "short", "medium", and "long" education are confusing from an English-language perspective, and do not appear to be standard.

The references to other countries, and the impact on screening and followup there, are a useful addition to the discussion – however, the inclusion of specific quantitative data would strengthen the comparisons. In particular, it is unclear whether countries such as Spain and England have since ensured that those that missed screening returned after the formal pauses. My understanding is that they have not, which would strengthen the argument that the approach in Denmark was superior.

I feel that a greater emphasis on the potential downstream impact on colorectal cancer diagnoses and deaths would make the point of the paper clearer – for instance, de Jonge et al. "Impact of the COVID-19 pandemic on faecal immunochemical test-based colorectal cancer screening programmes in Australia, Canada, and the Netherlands: a comparative modelling study" and Mandrik et al. "Modelling the impact of the coronavirus pandemic on bowel cancer screening outcomes in England: A decision analysis to prepare for future screening disruption". Screening is of course a means to an end, rather than an end itself, so this would contextualise the results for the reader.

Due to the nature of the study, the novelty is relatively limited – this is a very straightforward statistical analysis of screening registry data, and the subgroup analysis reveals few surprises. Ideally, such analysis would be provided by the registry themselves. In the absence of this though, I believe that it is important that this work is available to the public and I recommend publication.

*Reviewer #2 (Recommendations for the authors):*

The paper does not need any important changes. If the editors think this paper is not interesting enough, I suggest making one paper about the three screening programs in Denmark, instead of several companion papers. This would allow a comparative discussion and would make the papers less descriptive.

I have very few comments, since the paper is very well written and the report is rigorous, the methods well documented, tables and figure clear.

Abstract

No comment

Introduction

Line101-103: it is not clear the link between the sentence and the topic of the paper. It should be explained why you are talking about health seeking behaviors.

Methods

In the exposure of interest and in the outcome, it could be explained that the periods classify the date of invitation that is not the date of the event. It is not clear if the compliance to colonoscopy is classified on the day of invitation for colonoscopy of for FIT.

Discussion

The discussion is too long. Many paragraphs are very speculative. I appreciate the effort to give an interpretation of the observed phenomena in the light of other studies, but in some cases there is no other evidence or insight supporting the interpretation that sounds more common sense than interpretation (lines 359-369–lines 401 411).

Lines 430-432: the authors did not provide any evidence that participation to screening was safe, I mean attending clinics for colonoscopies. They did not provide data about the risk of SARS-CoV-2 infection of people attending endoscopic services.

Comparisons with data from other screening programs are the most interesting comments in the discussion, maybe only one paper focusing on differences would be more intriguing.

---

## [Author Response]

Reviewer #1 (Recommendations for the authors):It would be useful to include a reference for the assertion that "most" programmes were paused – these references were included in the discussion but not in the introduction.

Thank you for this comment. Additional references have been added in the Introduction and Discussion section (page 3, line 80 and page 9, line 340).

The language around "short", "medium", and "long" education are confusing from an English-language perspective, and do not appear to be standard.

Thank you for this comment. We have used the International Standard Classification of Education (ISCED) of the United Nations Education, Scientific and Cultural Organization (UNESCO) for the categorisation of educational level. ISCED 1-2 comprise primary education to upper secondary education, ISCED 3-5 comprise vocational education and training to vocational bachelors educations and ISCED 6-8 comprise bachelors programmes to PhD programmes. We have added this explanation in the Methods section (page 5, lines 200-205).

The references to other countries, and the impact on screening and followup there, are a useful addition to the discussion – however, the inclusion of specific quantitative data would strengthen the comparisons. In particular, it is unclear whether countries such as Spain and England have since ensured that those that missed screening returned after the formal pauses. My understanding is that they have not, which would strengthen the argument that the approach in Denmark was superior.

Thank you for this comment. We have elaborated on this in the Discussion section (page 9, lines 345-360 and pages 9-10, lines 367-379).

I feel that a greater emphasis on the potential downstream impact on colorectal cancer diagnoses and deaths would make the point of the paper clearer – for instance, de Jonge et al. "Impact of the COVID-19 pandemic on faecal immunochemical test-based colorectal cancer screening programmes in Australia, Canada, and the Netherlands: a comparative modelling study" and Mandrik et al. "Modelling the impact of the coronavirus pandemic on bowel cancer screening outcomes in England: A decision analysis to prepare for future screening disruption". Screening is of course a means to an end, rather than an end itself, so this would contextualise the results for the reader.

Thank you for this comment. We have elaborated on the potential downstream effect of colorectal cancer screening and added the mentioned references in the Discussion section (page 10, lines 381-405).

Due to the nature of the study, the novelty is relatively limited – this is a very straightforward statistical analysis of screening registry data, and the subgroup analysis reveals few surprises. Ideally, such analysis would be provided by the registry themselves. In the absence of this though, I believe that it is important that this work is available to the public and I recommend publication.

Thank you for this comment. These analyses are not provided routinely by the clinical quality registries.

Reviewer #2 (Recommendations for the authors):The paper does not need any important changes. If the editors think this paper is not interesting enough, I suggest making one paper about the three screening programs in Denmark, instead of several companion papers. This would allow a comparative discussion and would make the papers less descriptive.I have very few comments, since the paper is very well written and the report is rigorous, the methods well documented, tables and figure clear.IntroductionLine101-103: it is not clear the link between the sentence and the topic of the paper. It should be explained why you are talking about health seeking behaviors.

Thank you for this comment. We have omitted this sentence.

MethodsIn the exposure of interest and in the outcome, it could be explained that the periods classify the date of invitation that is not the date of the event. It is not clear if the compliance to colonoscopy is classified on the day of invitation for colonoscopy of for FIT.

Thank you for this comment. We have clarified this in the Methods section (page 5, lines 181-182 and 188-190).

DiscussionThe discussion is too long. Many paragraphs are very speculative. I appreciate the effort to give an interpretation of the observed phenomena in the light of other studies, but in some cases there is no other evidence or insight supporting the interpretation that sounds more common sense than interpretation (lines 359-369–lines 401 411).

Thank you for this comment. We have shortened the Discussion section (pages 10-11, lines 407-439, page 11, lines 449-454 and 461-471 and page 12, lines 496-197).

Lines 430-432: the authors did not provide any evidence that participation to screening was safe, I mean attending clinics for colonoscopies. They did not provide data about the risk of SARS-CoV-2 infection of people attending endoscopic services.

Thank you for this comment. We have re-phrased this sentence (page 12, line 492).

Comparisons with data from other screening programs are the most interesting comments in the discussion, maybe only one paper focusing on differences would be more intriguing.

Additional changes:

We have omitted Supplementary Figure 2 and 3 (stratification by explanatory variables) as these data are already displayed in Supplementary Table 1 and 4.